# Exploring the Relationship between the Gut Mucosal Virome and Colorectal Cancer: Characteristics and Correlations

**DOI:** 10.3390/cancers15143555

**Published:** 2023-07-09

**Authors:** Gangping Li, Yu Jin, Baolong Chen, Aiqiang Lin, Erchuan Wang, Fenghua Xu, Gengcheng Hu, Chuanxing Xiao, Hongli Liu, Xiaohua Hou, Bangzhou Zhang, Jun Song

**Affiliations:** 1Division of Gastroenterology, Union Hospital, Tongji Medical College, Huazhong University of Science and Technology, Wuhan 430022, China; ligangping@hust.edu.cn (G.L.); jinjoey@hust.edu.cn (Y.J.); 2010xh1051@hust.edu.cn (E.W.); xufenghua999@163.com (F.X.); hugengcheng@sina.com (G.H.); houxh@hust.edu.cn (X.H.); 2Xiamen Treatgut Biotechnology Co., Ltd., Xiamen 361115, China; chenbaolong@treatgut.com (B.C.); linaiqiang@treatgut.com (A.L.); xiaoxx@treatgut.com (C.X.); 3Cancer Center, Union Hospital, Tongji Medical College, Huazhong University of Science and Technology, Wuhan 430022, China; hongli_liu@hust.edu.cn; 4Institute for Microbial Ecology, School of Medicine, Xiamen University, Xiamen 361102, China

**Keywords:** gut mucosal virome, colorectal cancer, eukaryotic viruses, *Anelloviridae*, *Torque teno virus*

## Abstract

**Simple Summary:**

This study is for the first time to analyze the mucosal virome in patients with colorectal cancer (CRC) compared to healthy controls and show the differences between them. Unlike the changes in fecal virome in CRC (mainly associated with bacteriophages), this study revealed that the difference in the mucosal virome between CRC and HCs was dominated by eukaryotic viruses other than bacteriophages. The enrichment of TTV, one sub-species of Anelloviridae, in CRC tissues suggests that they may play an important oncogenic role in CRC, which will pave the way for innovative therapeutic strategies blocking or reverting CRC pathogenesis in the future.

**Abstract:**

The fecal virome has been reported to be associated with CRC. However, little is known about the mucosal virome signature in CRC. This study aimed to determine the viral community within CRC tissues and their contributions to colorectal carcinogenesis. Colonic mucosal biopsies were harvested from patients with CRC (biopsies of both neoplasia and adjacent normal tissue (CRC-A)) and healthy controls (HC). The shot-gun metagenomic sequencing of virus-like particles (VLPs) was performed on the biopsies. Viral community, functional pathways, and their correlations to clinical data were analyzed. Fluorescence in situ hybridizations (FISH) for the localization of viruses in the intestine was performed, as well as quantitative PCR for the detection of Torque teno virus load in human mucosal VLP DNA. A greater number and proportion of core species were found in CRC tissues than in CRC-A and HC tissues. The diversity of the mucosal virome in CRC tissues was significantly increased compared to that in HC and CRC-A tissues. The mucosal virome signature of CRC tissues were significantly different from those of HC and CRC-A tissues at the species level. The abundances of eukaryotic viruses from the Anelloviridae family and its sub-species Torque teno virus (TTV) were significantly higher in CRC patients than in HC. Furthermore, increased levels of TTV in the intestinal lamina propria were found in the CRC group. Multiple viral functions of TTV associated with carcinogenesis were enriched in CRC tissues. We revealed for the first time that the mucosal virobiota signature of CRC is characterized by a higher diversity and more eukaryotic viruses. The enrichment of TTV species in CRC tissues suggests that they may play an oncogenic role in CRC. Targeting eukaryotic viruses in the gut may provide novel strategies for the prevention and treatment of CRC.

## 1. Introduction

A total of 2.2 million cancer cases were attributed to infection in 2018. Approximately 60% of new cancers are associated with viral infection [1]. Viruses are the most abundant and widely distributed biological entities in the intestine [2]. Permanent and latent viral infections affect the physiology and pathology of the intestine [3]. CRC is the third most diagnosed malignancy and the second leading cause of cancer-related death [4]. The development of CRC is influenced by both environmental and genetic factors [5]. Although the gut bacteria are the most frequently investigated microbiota in CRC-related studies [6], growing evidence has highlighted that dysbiosis of the gut virome is responsible for a direct or indirect effect on CRC. Establishing a direct link between the intestinal virome and CRC will pave the way for the development of innovative therapeutic strategies blocking or reverting CRC pathogenesis [7].

The human virome is a diverse and abundant collection of all viruses found in or on humans and includes both eukaryotic and prokaryotic viruses, encompassing both animal-infecting viruses and bacteriophages [8]. Bacteriophages populating the gut microbiota act as predators of bacterial species and maintain the bacterial species diversity of the intestinal tract through predator–prey relationships [9], which suggests an indirect effect on gastrointestinal health and physiology. Indeed, two studies reported that the fecal virome (including bacteriophages) plays an indirect role in impacting CRC by modulating the associated bacterial community [10,11]. However, this put a magnification lens on bacteriophages, inevitably leading to disregard eukaryotic-infecting viruses that were only identified by chance upon stool analysis and are therefore underestimated [7]. Multiple cancers are directly associated with viral infection by human papillomavirus, hepatitis B virus and hepatitis C virus [1]. Therefore, it is urgent to detect the viral signature in colorectal tissues which may represent a more direct association between the virus and development of CRC. Additionally, the most used method to detect a viral sequence within a sample is polymerase chain reaction (PCR), which creates a large hurdle for discriminating viral/phage abundance and diversity because of the limited available references on which primers are designed [12]. Metagenomics represents an unbiased approach to uncover the viral communities within a sample, but there is a lack of reports about CRC tissue-related viral signatures using this approach.

Here, we performed a study to reveal the gut mucosal virome alterations in CRC compared to HC using the metagenomic sequencing of VLPs. To our knowledge, this is the first study to characterize the virome of CRC tissues and to show the association of these virome compositions and functions with clinical characteristics. These data supplied direct information about potential oncogenic viruses in CRC.

## 2. Materials and Methods

### 2.1. Cohort Description and Study Participants

Twenty-four clinical primary diagnoses of CRC patients and 18 HCs who provided informed consent were included in this study. Participants who underwent colonoscopy with standard bowel preparation at the Endoscopy Center of Wuhan Union Hospital were recruited. The CRC diagnoses were confirmed by a pathologist. HC referred to participants who underwent colonoscopy and had no positive findings, such as polyps and inflammation. Patients who had consumed prebiotics, yogurt and antibiotic or antiviral drugs within 3 months were excluded. Patients who had a coexisting disease, such as inflammatory bowel disease, liver diseases, and pulmonary, cardiovascular or renal comorbidities were excluded. Demographic data such as age, sex, and stages of TNM and cancer differentiation were collected. Patient metadata are shown in Appendix A. Four biopsies of each subject were harvested from the tumor (CRC) and adjacent normal mucosa (approximately 5 cm from the tumor lesion, CRC-A) of CRC patients or from the normal mucosa of HC, and 3 biopsies were used for sequencing. Biopsies were collected via endoscopy and then stored at −80 °C until downstream mucosal virome analysis. Samples from 18 CRC patients and 15 HCs were included in virome analysis.

### 2.2. Mucosal VLP Enrichment and Sequencing

The VLPs of mucosal biopsies were enriched using a protocol modified from previously described methods [13]. Briefly, each mucosal biopsy was cut into small pieces with surgical scissors and digested in 1 mL digestion buffer (1 mg/mL collagenase D, 1 U baseline DNase I, PBS, pH 7.5) at 37 °C for 1 h, with intermittent intensive vertexing every 20 min. The biopsy suspension was then centrifuged at 5000× *g* for 5 min, and the supernatant was further passed through a 0.22 μm filter to remove debris and residual host and bacterial cells. To degrade and remove the remaining bacterial and host cell membranes, filtrate samples were treated with lysozyme (1 mg/mL at 37 °C for 30 min) followed by chloroform (0.2× volume at room temperature (RT) for 10 min) and centrifuged at 1500× *g* for 5 min. Non-VLP DNA in the supernatant was digested at 37 °C for 60 min by a DNase cocktail (14 U Turbo DNase (Invitrogen, Waltham, MA, USA), 250 U Benzonase Nuclease (Yeasen, Shanghai, China)) and 2 U RNase A (Sigma, St. Louis, MO, USA), followed by heat inactivation of nucleases at 65 °C for 10 min. VLP DNA was then extracted using the PureLink Viral RNA/DNA Mini Kit (Thermo Fisher Scientific, Waltham, MA, USA) according to the operating instructions. VLP DNA was amplified for 4 h using the Discover-sc Single Cell Kit V2 Kit (Vazyme, Nanjing, China) following the instructions provided by the manufacturer and quantified by a Qubit 3.0 fluorimeter (Invitrogen, Waltham, MA, USA). Both VLP DNAs and the negative control (Nuclease-free ddH_2_O) were amplified for 4 h by multiple displacement amplification (MDA) and were detected by agarose gel electrophoresis; our data (not shown) showed that all samples had obvious dispersion bands with a size of about 400 bp–1000 bp, while the negative control did not have a such similar band, except for the weak primer dimer at the bottom of the electrophoretic lane.

DNA samples (200 ng) were randomly fragmented into 350 bp fragments with an Ultrasonic Breaker Bioruptor NGS Sonicator (Diagenode, Liège, Belgium) and subjected to library preparation with an NEBNext^®^ Ultra™ II DNA Library Prep Kit for Illumina according to the manufacturer’s instructions. The qualified library was sequenced with 151 bp read chemistry in a paired-end flow cell on an Illumina NovaSeq platform (Illumina, Inc., San Diego, CA, USA).

### 2.3. Read Assembly and Taxonomic Assignment and Bioinformatics Analysis

Raw paired-end reads were quality-filtered and analyzed using an in-house pipeline that mainly included three steps. First, raw reads were filtered by Trimmomatic to remove low-quality reads and adaptors [14]. High-quality reads contaminated from the human host were removed from the dataset by mapping reads to the human sequence database (hg19) using KneadData 15. Finally, the remaining reads were taxonomically classified using Kraken2 by mapping to a reference database that contains archaea, bacteria, fungi, and viruses [15]. Nonviral taxa were removed from the taxonomy table, and all samples were resampled to an even sequencing depth for downstream analyses. Additionally, to assess the functions of the virome, all viral sequences were retrieved and assembled into contigs by Megahit [16]. Open reading frames were predicted from contigs using MetaGeneMark [17]. Kyoto Encyclopedia of Genes and Genomes (KEGG)-pathway enrichment analysis of the genes was implemented using KOBAS software 2.0with the default parameters [18].

### 2.4. Quantitative PCR for Detection of Torque Teno Virus Load in Human Mucosal VLPs DNA

TTV DNA-load quantification was carried out with a TaqMan real-time PCR targeting a highly conserved segment of the untranslated region (UTR) of the viral genome [19]. The following primers and probe were used for PCR amplification [20]: forward primer 5′-GTGCCGIAGGTGAGTTTA-3′, reverse primer 5′-AGCCCGGCCAGTCC-3′, probe 5′-TCAAGGGGCAATTCGGGCT-3′. The PCR amplification was performed with the Applied Biosystems 7300 Plus Real-Time PCR System (Applied Biosystems, Waltham, MA, USA) using the following reaction conditions: 50 °C for 2 min, 95 °C for 1 min, followed by 40 cycles of 95 °C of 15 s and 60 °C of 60 s. The cycle threshold (Ct) values for each sample were used for subsequent analysis. The Ct value is defined by the number of cycles in qPCR required for the fluorescent signal to cross the threshold.

### 2.5. In Situ Hybridization of TT Virus-DNA in the Intestine

Digoxigenin-dUTP-labeled DNA probe was designed according to the TT virus sequence in GenBank: AY823988.1, and the sequence of primer as follows: 5′-ACCAGAAGCACCACAACACGTGGAGCTATCCCAACAACCA-3′; 5′-ATTGGCAACAGAGATGGTGGCCCAGATTCAGCTTTCAGAG-3′. Freshly dissected tissue was fixed (<3 mm thick) with 2% paraformaldehyde from 1 h to overnight at room temperature. The tissue was rinsed with running tap water and the tissue was dehydrated. The tissue was cleared in xylene and immersed in paraffin, and the paraffin-embedded tissue block was sectioned at 4 μm thickness on a microtome and float in a water bath containing distilled water. The sections were transferred onto glass slides suitable for immunohistochemistry. The slides were dried overnight and stored at room temperature until ready for use. Deparaffinized slides in xylene were blocked endogenous peroxidase activity by incubating sections in 3% H_2_O_2_ solution in methanol at room temperature. Sections were digested by adding 3% citric acid freshly diluted pepsin (1 mL of 3% citric acid plus 2 drops of concentrated pepsin, mixed) at 37 °C temperature for 10 min, then in situ hybridization with PBS was performed. A volume of 100 μL blocking buffer (e.g., 10% fetal bovine serum in PBS) was added onto the sections of the slides and incubated in a humidified chamber at room temperature for 1 h. A volume of 20 μL digoxigenin-dUTP-labeled DNA probe was applied to the sections on the slides and incubated in a humidified chamber at room temperature for 1 h, and 100 μL appropriately diluted biotinylated secondary antibody (using the antibody dilution buffer) was applied to the sections on the slides and incubated in a humidified chamber at room temperature for 30 min. A volume of 100 μL appropriately diluted Biotinylated Anti-Digoxin was applied to the sections on the slides and incubated in a humidified chamber at room temperature for 30 min. The slides were washed in PBS for 5 min, four times, then incubated with SABC-FITC at 37 °C temperature for 60 min. The nucleus with DAPI was re-stained and the color of the antibody staining in the tissue sections was observed under microscopy.

### 2.6. Statistical Analysis and Visualization

Viral alpha diversity including Shannon and Pielou’s evenness (J’) was calculated using the R package vegan (v2.5.7). The significance of the differences measured in α-diversity metrics among groups was tested using a nonparametric Kruskal–Wallis rank sum test and Benjamini–Hochberg corrections with the R package agricolae (v1.3.3). Differences in community structure (β-diversity) were visualized using NMDS based on Bray–Curtis distances. The significance was determined by PERMANOVA with 999 permutations using adonis in the R package vegan. Differences in viral taxa and KEGG pathways were also tested using the Kruskal–Wallis rank sum test and Benjamini–Hochbery corrections with the R package agricolae. A *p*-value less than 0.05 as statistically different. The results were visualized using the custom R script mainly based on ggplot2 (v3.3.3). All analyses were performed using R v3.4.1.

## 3. Results

### 3.1. Overall Data from the Three Groups

The median ages of the CRC patients and HC were 57 and 51, respectively. A proportion of 60% of participants in the HC group were female while 66.7% of participants in the CRC group were female. Age and gender were comparable between CRC patients and HC (all *p* > 0.05, Table 1). 

Previous data showed that the intestinal virome is individual-specific. Herein, we accessed the distribution of viral species in all samples of different groups. More unique viral species which were shared by less than 20% of samples were detected in HC and CRC-A tissues but not in CRC tissues (Figure 1a). In terms of common viral species which were shared in 20–50% of samples, the HC group had the highest numbers of species, followed by CRC-A, while the CRC group had the smallest number of species (Figure 1a). However, the opposite trend was shown in the core species which was detected in at least 50% of all samples; more species and a higher proportion of core species were found in CRC tissues than in the HC and CRC-A (Figure 1b). The intestinal virome in HC was individual-specific but the viromes in CRC tissues had a similar signature. These data indicate that CRC patients share a unique mucosal virome characteristic.

### 3.2. Increased Viral Diversity and Richness in CRC Tissues

Viral diversity was used to show the distribution, relative abundance, and interindividual differences in viruses on the biodiversity dimension in different ecosystems. According to the above data, CRC exhibited a virome signature with more core species and fewer unique species than the HCs and CRC-A tissues. Then, we evaluated the differences in biodiversity among these three groups using Shannon diversity. We observed an increasing trend in diversity from the HC to CRC (*p* = 0.013, compared to HC) (Figure 2a). We also separately analyzed the diversity of bacteriophages and found no significant alterations in Shannon diversity among these three groups (all *p* > 0.05) (Figure 2b). These data suggest that diversity of non-bacteriophages other than bacteriophages in CRC patients increased.

For high individual specificity in the human gut virome, we compared the distribution of species among the three groups. The CRC group showed a higher evenness index than the HC and CRC-A groups (*p* = 0.0018 and *p* = 0.016) (Figure 2c). There were no significant differences in the evenness index between the HC and CRC-A groups (*p* = 0.4). Similarly, in terms of bacteriophages, there were no significant alterations in the evenness index among these three groups (all *p* > 0.05) (Figure 2d). These data indicate that alteration in the number of non-bacteriophage viruses decreased the variation of the distribution in CRC.

### 3.3. Alteration in the Mucosal Virome of CRC Tissues

Beta diversity metrics were used to capture viral community differences among the three groups. The data showed that the mucosal viromes of CRC tissues were significantly different from those of the HC and CRC-A tissues at the species level (based on Bray–Curtis dissimilarities, ANOSIM test, *p* < 0.001, *p* = 0.0039, Figure 3a), while no difference was found between HC and CRC-A tissues (ANOSIM test, *p* = 0.22). We also measured the difference in detectable bacteriophage species among HC, CRC-A and CRC tissues (Figure 3b), but we could not separate the CRC community from the HC and CRC-A communities by mucosal bacteriophage species (*p* = 0.787, Figure 3b). There was the same trend in the PCA analysis based on the whole virome, non-bacteriophage and bacteriophage among these three groups (Figure 3c–e). To determine the contribution of mucosal bacteriophage and non-bacteriophage species in the discrimination of different viral communities, we evaluated the contributions of bacteriophages and non-bacteriophages (eukaryotic viruses and other viruses) (Appendix A). The contribution of Anelloviridae was higher than that of phages (49.7% vs. 47%). These data suggest that mucosal virome dysbiosis in CRC was dominated by non-bacteriophages.

### 3.4. Distinct Community of Mucosal Virome in CRC

The fecal virome was mainly composed by bacteriophages, but little is known about mucosal samples. Viral taxa were compared among the HC, CRC-A and CRC groups at the family, genus and species levels. The top three most common viruses at the family level in these three groups were Podoviridae, Anelloviridae and Siphoviridae. The dominant virus (abundance > 1%) in HC was Podoviridae, while Anelloviridae was prevalent in CRC patients (in both the CRC-A and CRC groups, see Appendix A). At the genus level, Alphatorquevirus was the dominant virus in the three groups with different ratios (HC vs. CRC-A vs. CRC: 34% vs. 74% vs. 85%) (Appendix A). Uncultured crAssphage was the dominant viral species in the HC and CRC-A groups, while Torque teno virus 22 was dominant in the CRC group. These data show that bacteriophages were the dominant viruses in the HC and CRC-A groups at the family, genus and species levels, while eukaryotic viruses were prevalent in CRC (Appendix A).

At the family level, compared to the HC group, the CRC group showed decreased levels of Podoviridae, Myoviridae, Retroviridae and Poxviridae and increased levels of Anelloviridae (*p* < 0.001) (Figure 4a). At the genus level, Zetatorquevirus and Alphatorquevirus abundances were significantly increased in CRC (*p* < 0.001), while Efquatrovirus, Eneladusvirus and Lubbockvirus, which consist of bacteriophages, had a decreased abundance in CRC (Figure 4b). At the species level, the abundances of uncultured crAssphage, Streptococcus phage EJ 1 and Clostridium phage phiCT9441A were decreased in the CRC group, while the abundances of Torque teno virus 1, Torque teno virus 3, Torque teno virus 5, Torque teno virus 8, Torque teno virus 10, Torque teno virus 11, Torque teno virus 12, Torque teno virus 13, Torque teno virus 15, Torque teno virus 16, Torque teno virus 18, Torque teno virus 19, Torque teno virus 20, Torque teno virus 21, Torque teno virus 22, Torque teno virus 27, and Torque teno virus 29 increased (Figure 4c). The positive rate of Hapillomavirus, Herpesvirus, Polyomavirus and Hepatitis virus were detected in all samples, and data are shown in Appendix A. Collectively, the number of eukaryotic viruses increased in CRC compared to that in HC, while the bacteriophage abundance decreased at different levels. These data showed that the mucosal virome was dominated by eukaryotic viruses, especially by the Anelloviridae.

### 3.5. Increased Loads of Torque Teno Virus in CRC Tissue

Torque teno virus was prevalent in HC and CRC, and was closely related with CRC based on metagenomic data. We further investigated the Torque teno virus DNA load which was detected by TaqMan real-time PCR among the three groups. More TTV DNA loads were detected in CRC patients compared to HC patients (*p* < 0.0001, *p* = 0.013), and higher TTV loads were measured in CRC than CRC-A (*p* = 0.00029) (Figure 5a). The increased TTV DNA loads in CRC was consistent with what we found based on metagenomic data. We applied FISH to show the TTV in intestine tissue. The signals were observed in all samples from HC, CRC-A and CRC, and were mostly seen in intestinal lamina propria. The signal intensity in the CRC sample was higher than that in HC and CRC-A (Figure 5b).

We also evaluated the influence of TTVs between CRC stages and found that TTVs increased in the early stage of CRC. Specifically, Torque teno virus 15, Torque teno virus 27, Torque teno virus 28, and Torque teno virus 19 were enriched in stages I and II of CRC but decreased in stages III and IV (Appendix A).

### 3.6. Altered Virome Function in CRC Tissue

To assess the functions of the mucosal virome, KEGG pathway enrichment analysis of the genes was performed. More abundant molecular functions, such as adhesion function, glycerophospholipid metabolism, the phosphatidylinositol signaling system, and primary bile acid biosynthesis, were identified in the CRC group but not in the HC group. The abundance of the one-carbon pool by folate, base excision repair, and the cell cycle of Caulobacter were decreased in CRC (Figure 6a). Except for the cell cycle of Caulobacter, which is related to bacteriophages, all altered functions were linked to a direct impact on tumors. We then evaluated the relationship between viral taxa and functional genes, and data showed that TTVs were positively related with functional genes which might have promoted colorectal carcinogenesis, while viruses belonging to bacteriophages were related to functional genes prohibiting carcinogenesis (Figure 6b). These data suggested TTVs might promote the colorectal carcinogenesis through increasing the oncogenic mechanisms of the virus.

## 4. Discussion

To our knowledge, this is the first mucosal virome study to reveal the viral community alterations in patients with CRC using a dedicated metagenomics approach based on enriched virome preparation. Our data showed that CRC exhibited dysbiotic mucosal virobiota characterized by an increase in eukaryotic viruses and a decrease in bacteriophages, particularly in the tumor area. The abundance of Anelloviridae increased dramatically in CRC patients, and a few TTV species were related to the TNM stage of CRC. Collectively, our data showed that eukaryotic viruses (mainly Anelloviridae) other than bacteriophages were significantly altered in specific sites in CRC tissues, and we hypothesize that these viruses might play a direct role in promoting colorectal carcinogenesis.

Many works on the relationship between the gut microbiome and CRC were based on fecal samples for easy access. However, the fecal microbiota does not represent the mucosal microbial community [21]. This might also be the case for the gut virome [22]. Previous data have demonstrated that the fecal virome is large and diverse, stabilizes in adulthood, and is highly personalized [23]. Our data also showed highly individual differences in the mucosal virome in healthy participants; however, we found that the intratumoral virome exhibited more shared species in CRC patients than in HCs and CRC-A intestinal tissue, which suggests a unique intratumoral viral signature in CRC. We found that the diversity of the gut mucosal virome was higher in CRC patients than in HCs, which was consistent with previous study that reported an increased diversity of the fecal virome in CRC patients [11]. However, it is noteworthy that the diversity of the gut bacteriophage community in mucosa was not significantly different in CRC tissues compared to HC tissues. Since the gut virome is mainly composed of eukaryotic viruses and bacteriophages [24], these results indicate that eukaryotic viruses had a greater contribution to the increased diversity in CRC tissues. We also found that the viral community in CRC patients significantly differed from that in HC, while we failed to observe this difference based on bacteriophages. These data indicate that eukaryotic viruses other than bacteriophages had a specific role in CRC, which was different from what had been reported in the fecal virome [11]. Another animal study also showed an increase in certain bacteriophages’ abundance with tumorigenesis [25], but their data came from the fecal virome, which results in different findings. Additionally, the viral community of CRC-A was different from that of CRC tissues, but not from that of HC tissues, which indicates that locally altered viruses in the mucosa might participate in CRC carcinogenesis. These results also show that mucosal viruses might have a direct impact on CRC carcinogenesis, which coincides with the nature of eukaryotic viruses that affect host cell fate by directly interacting with the host [26].

The role of eukaryotic viruses populated in the gut mucosa was neglected due to the limitation of the detection technique [27]. Thanks to recently developed metagenomic sequencing and computational approaches, we found that eukaryotic viruses were dominant in CRC patients, while bacteriophages were dominant in HCs. Permanent and latent viral infections affect the physiology and pathology of the intestine [3]. We found Anelloviridae was a dominant family within the gut eukaryotic viromes [28,29] and was correlated with CRC. Anelloviridae species were previously detected in samples from one infant at different times [30], indicating early and persistent infections, which implicated the possibility of tumorigenesis as a cocarcinogen. Our data demonstrated that many subtypes of TTV that belong to the Anelloviridae family were enriched in the tumor (not on the surface of tumor) which suggested TTVs involved in CRC. TTV is known to infect various cell types and can be found throughout the body, and nearly two-thirds of the general population are infected with TTV. TTV is frequently found in many other biological fluids (saliva, feces, urine, genital secretions, tears, nasal fluid), thus suggesting that multiple methods of transmission exist. Preliminary studies suggest that there may be an association between TTV and the development of certain diseases such as liver-related diseases, immune-system disorders, lung diseases and neurological disorders. However, the exact mechanisms linking TTV to these diseases remain unclear and further studies are needed to elucidate them. Thus, to date, TTV and other AVs are regarded as “viruses waiting for a disease” or “orphans of illness” [31]. Although there was no convincing direct causality between TTV and any specific disease, Andrea Hettmann reported the prevalence of TTV in saliva and biopsy samples from head and neck cancer patients, which suggests that TTV might act as a cocarcinogen in certain cancers [32]. These studies support the idea about the role of TTV as a cocarcinogen in CRC. More interestingly, Torque teno virus [13], Torque teno virus 18, Torque teno virus 27, and Torque teno virus 29 increased only in CRC tissues, not in CRC-A and HC. This alteration might reflect specific interactions between sub-species and CRC. Previous studies have reported that a few eukaryotic viruses, such as herpesviruses [33,34], papillomaviruses [35], and hepatitis B virus [36], are also related to CRC, but our data showed that these viruses were not populated in the mucosa. This heterogeneity may be due to the differences in methodology, geography and population [8]. Hence, we highlighted the role of TTV in CRC carcinogenesis.

Oncogenic mechanisms of viruses generally include disrupting cell-cycle progression, altering metabolic reprogramming, inhibiting DNA damage repair, evading cellular immune systems and shaping the bacterial community structure [8]. We observed the enrichment of viral adhesion genes in CRC which might be associated with viral invasiveness [37], and we observed the enrichment of TTV in the lamina propria of the tumor and a higher diversity in CRC. Lipid and bile-acid metabolism is involved in CRC [38,39], and we found that the functional genes related to phosphatidylinositol, glycerophospholipid and the primary bile-acid metabolism reprogramming of host cells increased in mucosal viruses in CRC, probably creating conditions for viral replication [40,41,42], which may contribute to increase the load of TTVs and promote colorectal carcinogenesis. The decreased function of base excision repair in mucosal viruses of CRC could alter host genome integrity during the dynamic interactions between viruses and hosts [43]. Caulobacter is a CRC-associated microbiota [44], and a decrease in functional gene targeting Caulobacter in CRC tissue caused a decline in bacteriophages, which control the quantity of CRC-associated microbiota and indirectly induce oncogenesis [8]. Here, we provided potential oncogenic mechanisms of mucosal viruses in CRC based on metagenomics analysis, and further investigations are required to determine these mechanisms in the future.

There are several limitations of our study. Firstly, the sample size was small in our study. A study using a bigger sample size should be conducted in the future. Secondly, although food habits might have a profound effect on the gut virome, dietary factors were not addressed in the experimental design. Thirdly, methods of isolating VLPs from human-associated sample matrices often utilize a combination of filtration and centrifugation techniques for the concentration of VLPs, followed by the elimination of contaminating cells and free nucleic acids, and so, we cannot rule out a possible DNA amplification bias and host-DNA contamination. The last shortcoming of this study was the low number of known viral genomes in public databases [45]. Improvements in sequencing and bioinformatics techniques are needed to further investigate this viral “dark matter”.

## 5. Conclusions

In conclusion, this is the first attempt to investigate the intestinal mucosal viral community associated with CRC by viral metagenomics and highlights the importance of the gut mucosal virome in CRC carcinogenesis. Mucosal virome in CRC is characterized by an increased biodiversity and an increase in eukaryotic viruses (not bacteriophages), which may influence host homeostasis and contribute to oncogenesis. The identified TTV species belonged to the Anelloviridae family and related functional genes provided new insights into CRC carcinogenesis. Targeting eukaryotic viruses (such as TTV species) in the gut may provide novel strategies for the prevention and treatment of CRC.

## Figures and Tables

**Figure 1 cancers-15-03555-f001:**
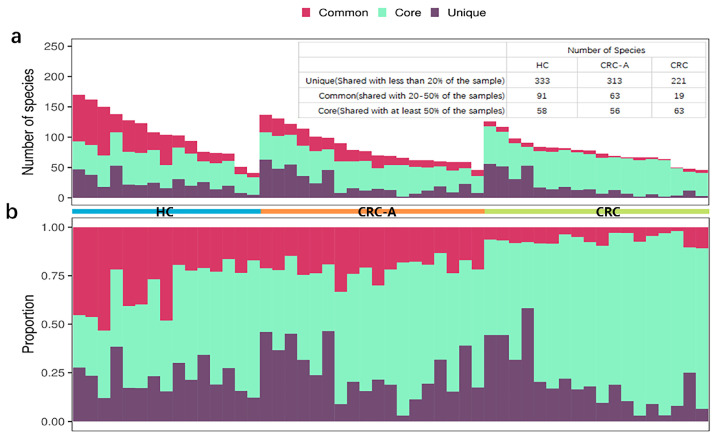
Distribution of shared viral species in HC, CRC-A and CRC group. Core species (detected in at least 50% of the samples), common species (shared in 20–50% of the samples) and unique species (detected in equal or less than 20% of the samples) are shown. In panel (**a**,**b**), one column represents one patient. (**a**). Absolute number. (**b**). Relative number of detected species in relationship to sharing characteristics in three different groups.

**Figure 2 cancers-15-03555-f002:**
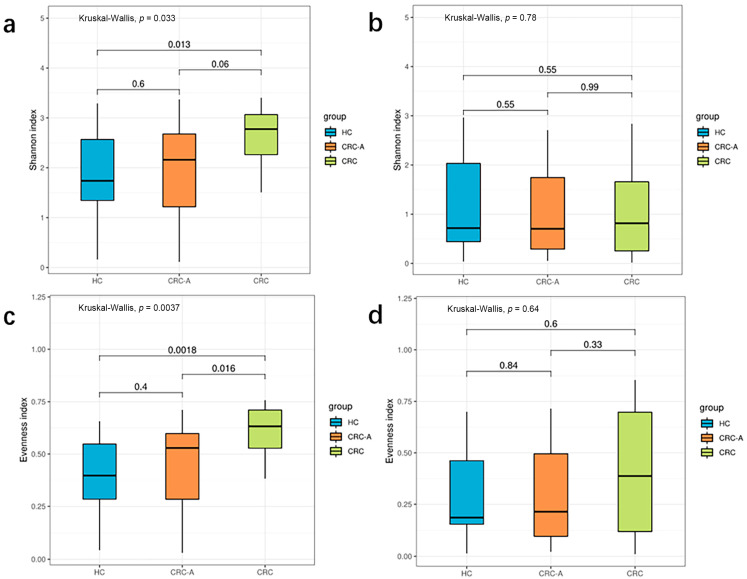
Biodiversity of the whole viral community and phage in HCs and CRC patients. (**a**) Shannon index of whole viral community in three groups. (**b**) Shannon index of phages in three groups. (**c**) Evenness of whole viral community in three groups. (**d**) Evenness of phages in three groups (*p* values were FDR corrected).

**Figure 3 cancers-15-03555-f003:**
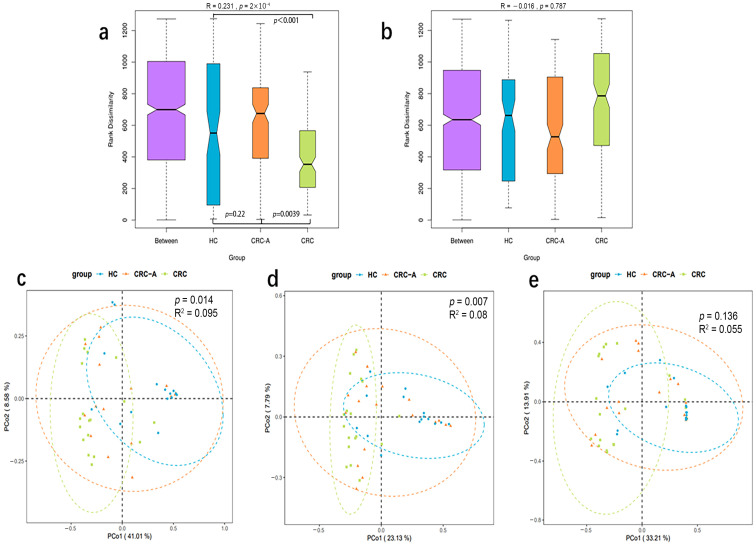
Beta diversity of the whole viral community, eukaryotic viruses and phages among HCs, CRC-A and CRC group. (**a**) ANOSIM analysis of the whole viral community among three groups. (**b**) ANOSIM analysis of the phages among three groups. (**c**) PCA analysis of the whole viral community among three groups. (**d**) PCA analysis of eukaryotic viruses in three groups. (**e**) PCA analysis of the phages among three groups (*p* values were FDR corrected).

**Figure 4 cancers-15-03555-f004:**
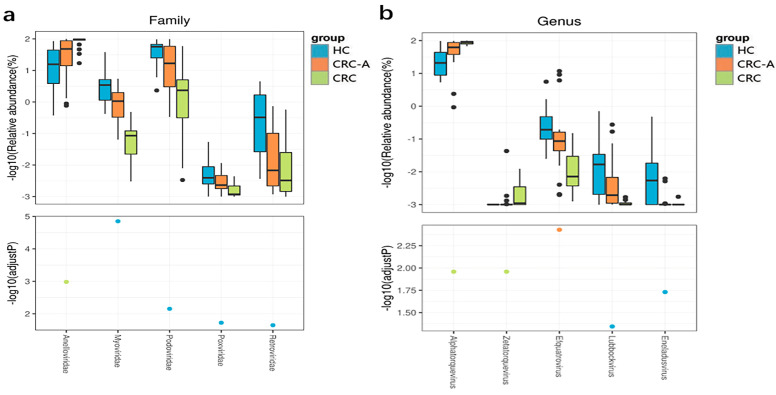
Altered mucosal viral composition in patients with CRC at different levels compared to HCs. Different viral taxa among HCs, CRC-A and CRC at the family (**a**), genus (**b**) and species levels (**c**). (*p* values were FDR corrected). The black dots represent discrete samples. Green and blue dots indicate different groups of *p*-values.

**Figure 5 cancers-15-03555-f005:**
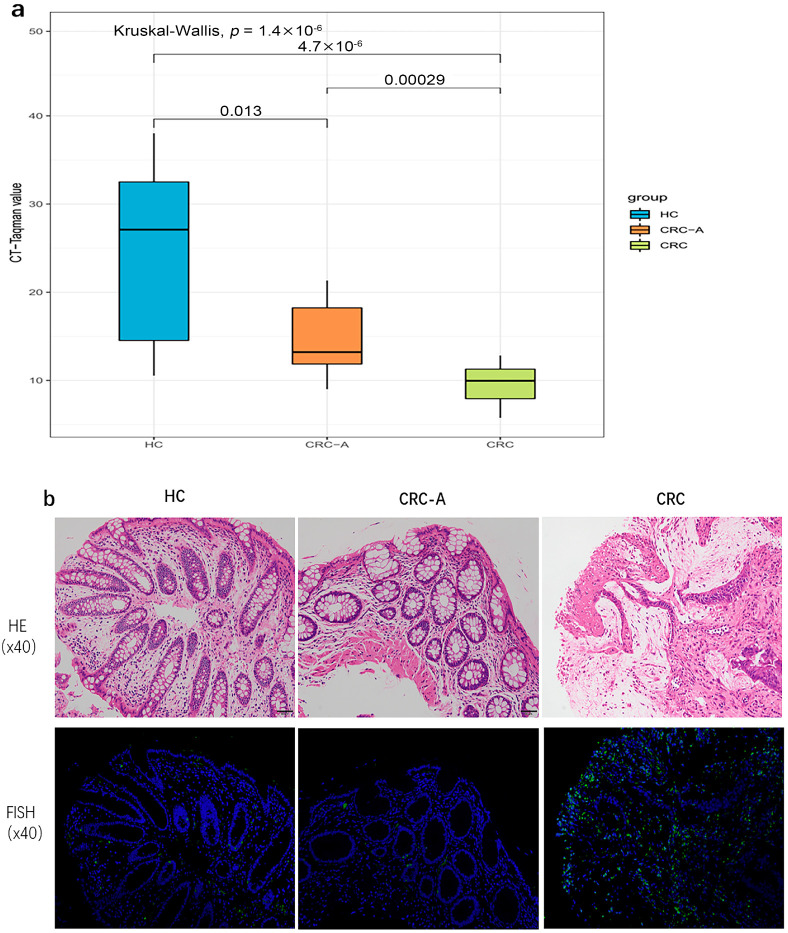
Torque teno virus DNA loads and location in HC and CRC tissue. (**a**) The cycle threshold (Ct) values, defined by the number of cycles in qPCR required for the fluorescent signal to cross the threshold for each sample. (**b**). In situ hybridization of TT virus (TTV) DNA shows positive signals in the intestinal lamina propria.

**Figure 6 cancers-15-03555-f006:**
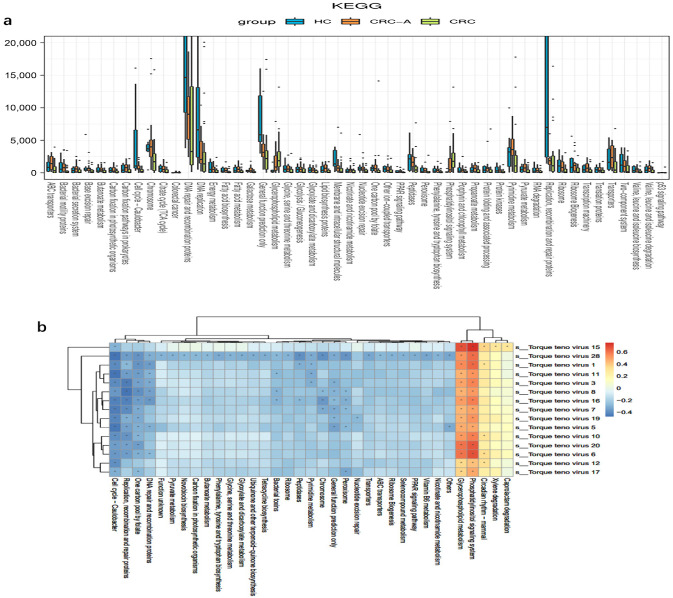
Functional alterations of mucosal virome in CRC. (**a**) Differentially enriched viral functions between HCs and CRC groups. Differential viral functions were determined by DESeq analysis with FDR correction. Only those functions with adjusted *p* < 0.05 and Log2 (between-group fold change) > 2 are shown. (**b**) Heatmaps show color-coded Spearman correlations of TTVs and predicted functions. Red color indicates positive correlation and blue indicates negative correlation (* *p* < 0.05).

**Table 1 cancers-15-03555-t001:** Characteristics of participants.

	HC	CRC	*p*
Age, years	51 (31–67)	57 (49–79)	>0.05
Gender female, n (%)	11 (61.1%)	16 (66.7%)	>0.05
Size of tumor	-	18.08 cm^2^	-
Stage of TNM, Early CRC (stage I & II), n%	-	12 (50%)	-
Differentiation	-		-
Well & Moderate	-	19	-
Poor	-	5	-

## Data Availability

The raw sequence data reported in this paper have been deposited in the Genome Sequence Archive (Genomics, Proteomics & Bioinformatics 2017) in National Genomics Data Center (Nucleic Acids Res 2021), China National Center for Bioinformation/Beijing Institute of Genomics, Chinese Academy of Sciences, under accession number CRA004054, and are publicly accessible at https://bigd.big.ac.cn/gsa (accessed on 1 April 2022).

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
