# Peer review of "Exploring the Relationship between the Gut Mucosal Virome and Colorectal Cancer: Characteristics and Correlations"

_cancers, 2023, doi:10.3390/cancers15143555_

Round 1
Reviewer 1 Report
The study contributes valuable knowledge to the field of CRC research by shedding light on the mucosal virome signature and its potential implications in colorectal carcinogenesis. The findings suggest that the virobiota could be a promising target for future therapeutic interventions in CRC.
Overall, this paper is suitable for publication. However, the authors must address the following questions before it can be accepted:
- Based on this study, the authors propose a potential role of mucosal virome in the occurrence and progression of CRC. I would like to know the clinical implications of these findings for the prevention and treatment of CRC. Is it possible to develop virus-based therapeutic strategies to intervene in the progression of CRC?
- This study provides valuable information about the alterations in mucosal virome in CRC, are there any potential limitations that need to be discussed? The authors utilized mucosal VLP analysis. It would be helpful if they could provide more details on the advantages and limitations of current mucosal VLP methods. Additionally, it is unclear whether this method only detects viruses that are actively replicating within host cells. Finally, the authors should explain how they computationally distinguish virus sequences from endogenous retrovirus (ERV) sequences.
- There are many illustrations in the article that appear stretched when inserted. Please improve this in the revised version.
- “To assess the functions of the mucosal virome, KEGG pathway enrichment analysis of the genes was performed.” How did the authors perform the KEGG analysis? Which genes were used?
- The results of this study show significant differences in the mucosal virome between CRC tissues and healthy controls. I would like to know if these differences are consistent across different populations and geographical regions, and if there are other studies that support the findings of this study.
- Based on the findings of this study, I am curious about the future research directions and potential research questions. Are there any further experimental or clinical studies planned to validate these findings and gain a deeper understanding of the mechanisms involved?
Overall, the English is clear and understandable.
Author Response
- Based on this study, the authors propose a potential role of mucosal virome in the occurrence and progression of CRC. I would like to know the clinical implications of these findings for the prevention and treatment of CRC. Is it possible to develop virus-based therapeutic strategies to intervene in the progression of CRC?
Answers: We appreciated your questions on the clinical implications of our findings for the prevention and treatment of CRC. Several cancers have been found to be associated with chronic viral infections, such as liver cancer and hepatitis B virus, cervical cancer and HPV virus, and nasopharyngeal cancer and EBV. There are also many clinical successes in intervening with the virus through antiviral drugs and vaccines to prevent and control the development of these cancers. The most critical part of this process is to find the virus which is associated with the development of cancer. Our data showed a significant difference in mucosal-associated virome between CRC and HCs. Many eukaryotic viruses(especially, subtypes of TTV that belong to the Anelloviridae family)were enriched in tumor tissues. Although our current data do not show a causal relationship between TTV virus and colorectal carcinogenesis, they at least suggest that there is a relationship between them. Our next goal is to further expand the sample size to repeat what we found, and them verify whether there is a causal relationship between eukaryotic viruses (such as TTV virus) and colorectal carcinogenesis through animal and cellular experiments. If we could prove a causal relationship between one virus and colorectal carcinogenesis, an antiviral drug or vaccine against that virus would be a tremendous innovation in the prevention and treatment of colorectal cancer. We think our current findings are not yet of such great clinical application value, but, through our study, we suggest the possibility of eukaryotic viruses being involved in tumorigenesis and provide the role of enlightening ideas for future research in this area, which we think is the greatest significance of our current manuscript.
- This study provides valuable information about the alterations in mucosal virome in CRC, are there any potential limitations that need to be discussed? The authors utilized mucosal VLP analysis. It would be helpful if they could provide more details on the advantages and limitations of current mucosal VLP methods. Additionally, it is unclear whether this method only detects viruses that are actively replicating within host cells. Finally, the authors should explain how they computationally distinguish virus sequences from endogenous retrovirus (ERV) sequences.
Answers: We appreciate your constructive suggestions. We added a paragraph to discuss the limitation of our study in the discussion section, which included the advantages and limitations of current mucosal VLP methods. (Please see in revised manuscript with purple color).
According to the experimental methods described in the article, ‘ The biopsy suspension was then centrifuged at 5,000 x g for 5 min, and the supernatant was further passed through a 0.22 μm filter to remove debris and residual host and bacterial cells. To degrade and remove the remaining bacterial and host cell membranes, filtrate samples were treated with lysozyme (1 mg/ml at 37 ℃ for 30 min) followed by chloroform (0.2x volume at room temperature (RT) for 10 min) and centrifuged at 1500×g for 5 min. Non-VLP DNA in the supernatant was digested at 37°C for 60 min by a DNase cocktail (14 U Turbo DNase (Invitrogen), 250 U Benzonase Nuclease (Yeasen)) and 2 U RNase A (Sigma) ’, the most of the virus sequences from endogenous retrovirus (ERV) sequences may be removed. And there are currently approximately 30 known HERVs, including HERV-W and HERV-E, but none of them were obtained during the virus annotation.
- There are many illustrations in the article that appear stretched when inserted. Please improve this in the revised version. Answers: We apologize for the confusion and have revised them(please see the illustrations in the revised manuscript)
- “To assess the functions of the mucosal virome, KEGG pathway enrichment analysis of the genes was performed.” How did the authors perform the KEGG analysis? Which genes were used? Answers:We appreciate your constructive suggestions. First, we use the megahit to assemble the fastq file after quality control, and then use MetageneMark to perform Gene prediction on the assembled contigs file, while filtering out the nucleic acid sequence and protein sequence whose length is less than 100 bp. Afterward, the results from the previous step were de redundant analyzed using CD hit software. The results were then compared with the quality control Fastq file and the Kobas database to obtain the gene expression matrix and pathway results. Finally, customized code is used to associate genes with pathways and generate an expression pathway matrix. A total of 27882 genes were involved in functional enrichment analysis in the article, such as PLC, eutB, CKI1, pcs.
- The results of this study show significant differences in the mucosal virome between CRC tissues and healthy controls. I would like to know if these differences are consistent across different populations and geographical regions, and if there are other studies that support the findings of this study.
Answers: Thank you for your good question. According to a previous study, the human fecal virome is highly diverse, stable, and individual specific (Shkoporov AN, et al. The Human Gut Virome Is Highly Diverse, Stable, and Individual Specific. Cell Host Microbe. 2019 Oct 9;26(4):527-541.) and might be influenced by an antibiotic (Wang L, et al. Altered human gut virome in patients undergoing antibiotics therapy for Helicobacter pylori. Nat Commun. 2023 Apr 17;14(1):2196.). However, little is known about mucosal virome. The gut virome is comprised of eukaryotic and bacterial viruses of which phages make up the vast majority. Cross-assembly phages (CrAssphage) have been found as the prevalent constituent of the human gut viral population, possibly representing a new viral family of bacteriophages recently found to be present in thousands of human-feces-associated environments around the world. It is not yet known whether eukaryotic viruses are also so widespread around the world. Therefore, it’s hard to know if these differences between CRC and HCs are consistent across different populations and geographical regions. As we mentioned in the manuscript, this is the first study to characterize the mucosal virome of CRC, we have found no study to support the findings of our study directly. But more and more researchers focused on gut eukaryotic virome in colorectal carcinogenesis (Massimino L, et al. Gut eukaryotic virome in colorectal carcinogenesis: Is that a trigger? Comput Struct Biotechnol J. 2020 Dec 7;19:16-28. Emlet C, et al. Enteric Virome and Carcinogenesis in the Gut. Dig Dis Sci. 2020 Mar;65(3):852-864.). TT Virus Infection Is Widespread in the General Populations from Different Geographic Regions.
- Based on the findings of this study, I am curious about future research directions and potential research questions. Are there any further experimental or clinical studies planned to validate these findings and gain a deeper understanding of the mechanisms involved?
Answers: We appreciate your suggestions. Our next goal is to further expand the sample size to repeat what we found, and then verify whether there is a causal relationship between eukaryotic viruses (such as TTV virus) and colorectal carcinogenesis through animal and cellular experiments.

Reviewer 2 Report
The Authors provided the interesting results which appreciated the role of virome during the cancerogenesis. They concentrated on colorectal cancer which is the crucial clinical problem since it causes thousands cancer-related deaths each year.
The detailed analysis of viruses which may settle a gut and may exert significant impact on the functions of mucosa and the diversity of intestinal microbiota, is suggested to contribute to the development of novel therapies and preventive strategies.
The manuscript is written with good English, the Results part requires minor revision.
Some of the detailed remarks are presented below:
Title:
· in a current form, in my opinion, it is confusing. Try to improve it. I believe it can be better.
Introduction:
This paragraph presents most of information necessary to understand the goal of study and results. What I need more is the explanation concerning TTV (Torque teno virus), why it can be a serious clinical problem? Its presence in intestinal mucosa is often in general population? Can it cause any other symptoms? Not only associated with cancer? Can it be found in different organs? The readers need to know if TTV is commonly found infective particles or only locally in China? Is it something we need to be scared of? Authors can add information about these issues in Discussion.
Line 68 – the lack of full stop at the end of sentence.
Results:
It is more familiar for readers when subparagraphs would have some kind of introduction with the aim of given experimental part presenting the relations between causes and effects. Consider this suggestion.
The advantage is the unification of colors in all the figures. It really makes it more clear.
Lines 196-197 – The sentence is too long and confusing. You can divide it into two parts. Consider rewriting especially its second part.
Line 198 – “No significant differences were found in age or sex…” – this sentence seems to be confusing. Difference in sex? Rather – difference in the number of male/female patients. Consider rewriting.
Lines 199-200 – “More unique viral species (shared in less than 20% of samples) were detected in HC and CRC-A tissues than in CRC tissues” – grammatically incorrect. “
Fig. 1 – the names of groups are hardly visible in the figure what makes the figure confusing. Modify the figure to make all information clear enough.
Lines 221-222 – “These data suggest that non-bacteriophages contributed more to the increased diversity in CRC patients.” – confusing sentence. Diversity of what? Sentence should be modified.
Taking into consideration that Authors took a lot of hard work to conduct all experiments and write a manuscript, it can be considered for publication in Cancers after corrections.
The manuscript is written with good English.
Author Response
Point 1: Title:in a current form, in my opinion, it is confusing. Try to improve it. I believe it can be better.
Response 1: We apologize for the confusion and appreciate your constructive suggestions. We have revised them. Our new title is as follows: Exploring the Relationship between the Gut Mucosal Virome and Colorectal Cancer: Characteristics and Correlations
Point 2: Introduction:
This paragraph presents most of information necessary to understand the goal of study and results. What I need more is the explanation concerning TTV (Torque teno virus), why it can be a serious clinical problem? Its presence in intestinal mucosa is often in general population? Can it cause any other symptoms? Not only associated with cancer? Can it be found in different organs? The readers need to know if TTV is commonly found infective particles or only locally in China? Is it something we need to be scared of? Authors can add information about these issues in Discussion.
Response 2: We appreciate your good suggestion. We added some information about TTV in the discussion. In short, nearly two-thirds of the general population are infected with TTV. TTV is known to infect various cell types and can be found throughout the body. TTV is frequently found in many other biological fluids (saliva, feces, urine, genital secretions, tears, nasal fluid), thus suggesting that multiple methods of transmission exist. Preliminary studies suggest that there may be an association between TTV and the development of certain diseases such as liver-related diseases, immune system disorders, lung diseases, and neurological disorders. However, the exact mechanisms linking TTV to these diseases remain unclear and further studies are needed to elucidate them. Thus, to date, TTV and other AV are regarded as "viruses waiting for a disease" or "orphans of illness".
Point 3: Line 68 – the lack of full stop at the end of sentence.
Response 3:We apologize for it and have revised them(see in revised manuscript with red color).
Point 4: Results:It is more familiar for readers when subparagraphs would have some kind of introduction with the aim of given experimental part presenting the relations between causes and effects. Consider this suggestion.
Response 4: We appreciate your good suggestions. We add some introduction in the front of and last sentence in each subparagraph. (Please see in revised manuscript with red color)
Point 5: Lines 196-197 – The sentence is too long and confusing. You can divide it into two parts. Consider rewriting especially its second part.
Response 5: We appreciate your good suggestions. We divided it into two parts. (Please see in revised manuscript with red color)
Point 6: Line 198 – “No significant differences were found in age or sex…” – this sentence seems to be confusing. Difference in sex? Rather – difference in the number of male/female patients. Consider rewriting.
Response 6: We appreciate your good suggestions. We rewrote it. (Please see in revised manuscript with red color)
Point 7: Lines 199-200 – “More unique viral species (shared in less than 20% of samples) were detected in HC and CRC-A tissues than in CRC tissues” – grammatically incorrect. “
Response 7:We appreciate your good suggestions. We corrected it. (Please see in revised manuscript with red color)
Point 8: Fig. 1 – the names of groups are hardly visible in the figure what makes the figure confusing. Modify the figure to make all information clear enough.
Response 8:We appreciate your good suggestions. We corrected it. (Please see in revised manuscript with red color)
Point 9: Lines 221-222 – “These data suggest that non-bacteriophages contributed more to the increased diversity in CRC patients.” – confusing sentence. Diversity of what? Sentence should be modified.
Response 9:We appreciate your good suggestions. We corrected it. (Please see in revised manuscript with red color)

Reviewer 3 Report
Authors have done a good study in comparing gut virome in the healthy and colorectal cancer patients. Though there are indications and predictions on presence of eukaryotic virome in such gut microenvironment, this is a first patient-based study and thus have critical value and importance. However, there are few concerns to be addressed before publishing in the journal.
As can be seen (Fig4C), some species of TTVs are more abundant in CRC but not in CRC-A which the authors propose is the evidence that these viruses are involved in Carcinogenesis, however, it is possible that the abundance could be merely due the tumor microenvironment which favors their colonization in the CRC rather than having anything to do with the promotion of Carcinogenesis.
As gender, age, and food habits can have a profound effect on the gut microbiome, including virome, these confounding factors should be considered while performing the statistical analysis.
It is not mentioned if the patients were receiving any chemotherapy or any other type of anti-cancer treatment which again can have an impact on the virome profile.
The authors have included a total of 24 CRC patients and 18 HCs in their study, however, only 18 patients and 15 HCs were included in the virome analysis without specifying the reason for the exclusion of the remaining subjects. The major limitation of this study is the small sample size. Any population/patient-based study requires a large sample size in order to reach a definitive conclusion.
As the authors have mentioned that 4-5 biopsy samples were collected from each patient but they have not mentioned whether all the biopsy samples were included in the analyses and whether the data is an average of both the individual biopsy samples and each individual patient.
The authors have observed a decrease in the viral load at higher stages of CRC yet they have not explained the reason for the same.
In the introduction: “Currently, approximately 60% of new cancers are associated with viral infection [1]”. Actually, the 60% is not mentioned in this reference. Please correct the figures or reference.
A similar but in vivo study in Murine model is reported previously (Fecal DNA Virome Is Associated with the Development of Colorectal Neoplasia in a Murine Model of Colorectal Cancer, Li et al, Pathogens. 2022; 11(4): 457). Please cite this and add a discussion.
There are many typo error and incomplete sentences.
For example on line no. 296 there are two p values given, without any explanation.
Author Response
Point 1:As can be seen (Fig4C), some species of TTVs are more abundant in CRC but not in CRC-A which the authors propose is the evidence that these viruses are involved in Carcinogenesis, however, it is possible that the abundance could be merely due the tumor microenvironment which favors their colonization in the CRC rather than having anything to do with the promotion of Carcinogenesis.
Response 1: We agree with your idea on this. Our data just show the relationship between the mucosal virus and CRC, and we assumed that some species of TTVs were involved in carcinogenesis. As we showed in Fig4C, some species of TTVs are more abundant both in CRC and CRC-A, which might be related to the whole changes in the gut. However, some species of TTVs are more abundant in CRC but not in CRC-A, which might just be related to the tumor itself. We are more concerned about the correlation between the virus and the tumor itself.
Point 2:As gender, age, and food habits can have a profound effect on the gut microbiome, including virome, these confounding factors should be considered while performing the statistical analysis.
Response 2: We totally agree with your suggestion. As we shown in results 3.1, we take age and gender into consideration, and age and gender were comparable between CRC patients and HC (all p > 0.05, Table 1). Patients who consumed prebiotics, yogurt, and antibiotic or antiviral drugs within 3 months were excluded. Patients who had a coexisting disease, such as inflammatory bowel disease; liver diseases; and pulmonary, cardiovascular, or renal comorbidities were excluded. Dietary habits are a difficult confounding factor to control because there is no way you can require all patients to eat the same diet, and it’s hard to pick patients in the clinical research.
Point 3:It is not mentioned if the patients were receiving any chemotherapy or any other type of anti-cancer treatment which again can have an impact on the virome profile.
Response 3: We totally agree with your suggestion. All the patients we recruited were primarily diagnosed with colorectal carcinoma and had not undergone any treatment. We add this to the method. (Please see in revised manuscript with green color)
Point 4:The authors have included a total of 24 CRC patients and 18 HCs in their study, however, only 18 patients and 15 HCs were included in the virome analysis without specifying the reason for the exclusion of the remaining subjects. The major limitation of this study is the small sample size. Any population/patient-based study requires a large sample size in order to reach a definitive conclusion.
Response 4:We appreciate your good suggestion. We totally agree with you that the major limitation of this study is the small sample size. We did not exclude any cases, and we used tissue specimens from 6 colorectal cancer cases, and 3 healthy controls were used for qPCR and situ hybridization.
Point 5:As the authors have mentioned that 4-5 biopsy samples were collected from each patient but they have not mentioned whether all the biopsy samples were included in the analyses and whether the data is an average of both the individual biopsy samples and each individual patient.
Response 5: We appreciate your good suggestion. We collected 4-5 biopsy samples, 3 biopsies were used for sequencing, and 1 was used for HE. DNA samples from biopsies were standardized before amplification. This statement might be misleading and was corrected. (Please see in revised manuscript with green color)
Point 6:The authors have observed a decrease in the viral load at higher stages of CRC, yet they have not explained the reason for the same.
Response 6: We appreciate your good suggestion. Yes, we did find that TTVs increased in the early stage of CRC. Specifically, Torque teno virus 15, Torque teno virus 27, Torque teno virus 28, and Torque teno virus 19 were enriched in stages I and II of CRC but decreased in stages III and IV. We just observed this phenomenon, but we didn’t know the reason. One possible reason is the tumor microenvironment which favors their colonization in the CRC changed in the later stage. This is an observed study and a deeper understanding of the mechanisms involved needs further experimental or clinical studies in the future.
Point 7:In the introduction: “Currently, approximately 60% of new cancers are associated with viral infection [1]”. Actually, the 60% is not mentioned in this reference. Please correct the figures or references.
Response 7:We apologize for the confusion and have revised them. (Please see in revised manuscript with green color). The original statement in [1] is: We found that, for 2018, an estimated 2·2 million infection-attributable cancer cases were diagnosed worldwide, corresponding to an infection-attributable ASIR of 25·0 cases per 100 000 person-years. Primary causes were Helicobacter pylori (810 000 cases, ASIR 8·7 cases per 100 000 person-years), human papillomavirus (690 000, 8·0), hepatitis B virus (360 00 million 0, 4·1) and hepatitis C virus (160 000, 1·7). The total number of viral infection-attributable cancer cases is 1.21 million (accounting for 55%, Approximately 60%).
Point 8:A similar but in vivo study in the Murine model is reported previously (Fecal DNA Virome Is Associated with the Development of Colorectal Neoplasia in a Murine Model of Colorectal Cancer, Li et al, Pathogens. 2022; 11(4): 457). Please cite this and add a discussion.
Response 8: We appreciate your good suggestion. We cited this and added a discussion in the manuscript. (Please see in revised manuscript with green color)
Point 9:Comments on the Quality of English Language
There are many typo error and incomplete sentences.
Response 9: We apologize for it and have revised them. (Please see in revised manuscript with green color)
Point 10:For example on line no. 296 there are two p values given, without any explanation.
Response 10: We apologize for it and have revised them. (Please see in revised manuscript with green color)
Round 2
Reviewer 1 Report
Thanks for addressing the concerns I raised in previous review. I am pleased to see that the authors have taken the time to carefully consider my comments and have made significant improvements to the work. The revisions the authors have made have effectively resolved the issues I had initially identified. I believe that the manuscript is now ready for publication.
Author Response
Thanks for addressing the concerns I raised in previous review. I am pleased to see that the authors have taken the time to carefully consider my comments and have made significant improvements to the work. The revisions the authors have made have effectively resolved the issues I had initially identified. I believe that the manuscript is now ready for publication.
Response: We sincerely appreciate your valuable comments and suggestions for revisions to my submitted manuscript. Your expertise and careful review have led to in-depth reflection and improvement of my research. Your suggestions during the review process have played a key role in improving the quality and accuracy of my paper. Thank you again for taking the time out of your busy schedule to review my paper.
Reviewer 3 Report
Authors have addressed most of the concerns. However, the concerns on the tumor microenvironment and dietary factors are not addressed in the experiment design. This is a limitation in this study and the authors have to mention it in the discussion.
Author Response
Authors have addressed most of the concerns. However, the concerns on the tumor microenvironment and dietary factors are not addressed in the experiment design. This is a limitation in this study and the authors have to mention it in the discussion.
Response: We sincerely appreciate your valuable comments and suggestions for revisions to my submitted manuscript. We added this limitation in the discussion section. (Please see in revised manuscript with green color.) Your expertise and careful review have led to in-depth reflection and improvement of my research. Your suggestions during the review process have played a key role in improving the quality and accuracy of my paper. Thank you again for taking the time out of your busy schedule to review my paper.
